# UV light and abrasion's role in degrading plasticulture films

**Ana Carolina Cugler Moreira** ⓘ **, Florian Wilken, Alessandro Fabrizi, Peter Fiener** ⓘ *

Institute of Geography, University of Augsburg, Augsburg, Germany

* peter.fiener@geo.uni-augsburg.de

## Abstract

Plastic films play a crucial role in agriculture by enhancing growing conditions and protecting crops through various applications, such as mulch films or greenhouses. However, the degradation of plasticulture films may occur in the field, generating secondary microplastics and soil pollution. Investigating the degradation of agricultural plastic films in soil systems is essential for understanding their environmental abundance, and it is crucial for developing effective mitigation strategies to avoid potential harm to soil health and agricultural productivity. In this study, various low-density polyethylene (LDPE) films commonly used in plasticulture were exposed to different times of ultraviolet (UV) radiation and to mechanical abrasion. Subsequently, the samples were analysed to assess their chemical properties, wettability, surface roughness, and elasticity changes. Although the films did not exceed their manufacturer's designated lifetime after the UV exposure time applied in this study, photodegradation and indications for the removal of microplastic surface particles through mechanical abrasion were observed even after short UV radiation exposure. Abrasion has been identified as an important modulator in photodegradation by removing UV radiation sheltering surface particles and thereby exposing undegraded film surfaces to UV radiation. However, changes in elasticity and susceptibility to fragmentation into macroplastic residues have not been identified.

## Introduction

The advent of plastic has revolutionised many sectors, including agriculture. The use of plastic in agriculture, known as plasticulture, started in the early 1950s with low-density polyethylene (LDPE) as mulch film and, since then, has been steadily increasing [1]. A large variety of plastics are used in agriculture, depending on the purposes, including different types of plastic films [2,3], irrigation systems, nets, transport, and packaging of agricultural food products and chemicals [4]. In modern horticulture, plastic films are used to cover the soil (i.e., mulch films) or the crop (i.e., tunnel and greenhouse films). Because of their crucial role, plastic films are the most commonly used plastic in vegetable production, with mulching films and greenhouse

**Data availability statement:** All data files are available from the Zenodo database (https://doi.org/10.5281/zenodo.14417456).

**Funding:** European Union's Horizon 2020 research and innovation programme under the Marie Skłodowska-Curie grant agreement No 955334.

**Competing interests:** The authors have declared that no competing interests exist.

accounting for $83 \times 10^6$ and $120 \times 10^6$ kg applied in Europe's agriculture, respectively [3]. Depending on the type of film, plasticulture can improve growing conditions by limiting evaporation, regulating temperature, increasing or decreasing sunlight absorption, suppressing weeds, or benefiting soil sterilisation [5,6]. To tailor them for their intended use, plastics are modified by chemical additives such as fillers, antioxidants, and stabilisers [5,7], affecting specific properties such as elasticity, strength, thermal and ultraviolet (UV)-radiation stability, and colour [5,6]. However, plastic film producers and vendors hardly provide any details regarding the use of additives. Plastic films are also produced in different thicknesses (8–300 μm), and can be separated into thin (< 80 μm) and thick (> 80 μm) films. Thin films are used as mulching for one growing season, while thick films are primarily used in horticulture, vegetable, or fruit production, where plastic films are used for multiple cropping seasons and are supposed to remain stable for a longer application time.

Despite the advantages introduced in agricultural production, weathering and mechanical stress may lead to the detachment of residues during mulch application or at the time of removal, especially when films approach the end of their intended lifespan [8,9]. This process can either lead to the formation of macroplastic (> 5 mm), which is subsequently fragmented into smaller pieces, particularly during tillage operations [10], or to the direct formation of secondary microplastic particles (< 5 mm) via abrasion [11]. Hann et al. [6] estimated that if at least 5% of the $83 \times 10^6$ kg of mulch film annually applied in Europe are not removed after use, $4.15 \times 10^6$ kg would remain in the soil system. Plastic residues have been shown to impact soil systems in terms of physical and chemical properties, with a pronounced effect on the soil biome, reducing plant growth and, hence, agricultural productivity [12,13]. Transport of plastic residues via wind [14], surface runoff, and soil erosion [15] connects agricultural plastic inputs with neighbouring ecosystems. Moreover, small microplastic particles may be transported vertically along the soil column [16] and hence result in the contamination of shallow groundwater [9,16]. Yang et al. [17] studied the microplastic formation from biodegradable, oxo-degradable, and non-degradable agricultural plastic films under different UV irradiation exposures in soil. Non-degradable films generated fewer microplastic particles, and crystallinity and abrasion were identified as key accelerating processes. Song et al. [18] found that LDPE pellets fragmented more under combined UV and abrasion exposure than with abrasion alone. Bhattacharjee et al. [19] also reported that non-agricultural LDPE films release more microplastics through abrasive forces and photodegradation exposure compared to pristine films. In contrast, Ren and Ni [20] questioned the primacy of photodegradation in microplastic formation of transparent agricultural LDPE film, suggesting wind-driven abrasion as a more significant contributor. To mitigate the plastic input to soil systems, improvements in the process of understanding the weathering and fragmentation pathways of different plastic films in agricultural environments are therefore essential [17,20,21].

This study aims to get a mechanistic understanding of photodegradation and follow-up abrasion using state-of-the-art analytical methods. The research questions of this study are: (i) What is the temporal development and degree of

photodegradation at different UV radiation exposure times for different types of films used in plasticulture? (ii) How does this photodegradation correspond to the formation of microplastics and macroplastics? (iii) What is the role of plastic film abrasion in degradation and for plastic input into soil systems?

## Materials and methods

### Experimental design

In this experiment, commercially available LDPE films were used, which are commonly applied as short-term (one season), long-term mulch films (multiple seasons), and greenhouse covers (multiple seasons). Two thin black mulch films (approx. 20 μm), two thick black and white mulch films (approx. 100–150 μm), and two thick transparent greenhouse covers (approx. 150–180 μm) were purchased at two different big farm suppliers in 2022 (Firmenich, Germany; BayWa AG Fürth, Germany) (Table 1).

For the experiment, an approximately 1 m long part of the mulch and greenhouse films were withdrawn from each specific roll. From each of the 1 m long samples, twenty 3 cm x 5 cm small samples (160 samples in total) were taken out. The samples were separated into 72 samples for different UV treatments, 72 samples for different UV plus abrasion treatments, 8 samples without UV but with abrasion treatment, and 8 samples representing pristine films as controls (Fig 1). Each 3 cm x 5 cm piece was placed on a magnetic frame with an internal window size of 2 cm x 4 cm and an external frame size of 4 cm x 5 cm to keep the samples stable and as flat as possible during the UV treatment.

For the UV treatment (Fig 1), a UV xenon test chamber (Q-SUN Xe-1-SE, Q-Lab, USA) equipped with one xenon arc lamp 1800 W (X-1800+, Q-Lab, USA), a Daylight-Q filter, and an insulated black panel temperature sensor was used. In terms of comparability between the artificial UV radiation source and natural sunlight exposure, the test chamber matches the ASTM G155 standard [22]. The UV test chamber was used with a spectral range from 300 to 400 nm (mainly UVA), irradiation of 75 $W \cdot m^{-2}$ (where this represents 6.48 $MJ \cdot m^{-2}$ for one day under UV exposure), and a chamber temperature of 50°C in a 24-hour light cycle. No other parameter was controlled. The time the samples spent in the chamber was set to simulate nine semi-randomised UV exposure times (21, 26, 31, 39, 40, 42, 52, 60, and 80 UV days), reflecting the variety of UV exposure conditions found in Europe. The samples were stored at 4°C between treatments and analysis to prevent further degradation.

Following the European organisation for technical approvals [23], an estimated recommendation of 201 MJ $m^{-2}$ was established for one year of equivalent radiation dose for the wavelength range between 300–400 nm. To determine the relative exposure of the samples to the natural conditions, the incident energy per day (6.48 $MJ \cdot m^{-2}$) should be multiplied by the number of exposure days and then divided by the one-year recommendation (201 $MJ \cdot m^{-2}$).

**Table 1. Main characteristics and typical low-density polyethylene (LDPE) film usage.**

| Abbreviation‡ | Measured thickness* (μm) | Type | Use | Duration use | Life span† | Colour |
|---|---|---|---|---|---|---|
| MS1 | 17.6±1.8 | thin mulch film on the soil surface | vegetables or fruits | one growing season | up to 36 months | black |
| MS2 | 18.4±1.0 | | | | | |
| ML1B ML1W | 98.9±2.7 | thick mulch film on the soil surface | asparagus | multiple growing seasons | up to 84 months | black/white |
| ML2B ML2W | 143.4±4.9 | | | | | |
| GL1 | 156.8±1.4 | large tunnel film for greenhouses | vegetables or fruits | multiple growing seasons | up to 54 months | transparent |
| GL2 | 185.3±5.9 | | | | | |

‡ MS1: mulch film, short term, no. 1; ML1B: mulch film, long term, no. 1, black side up; ML1W: mulch film, long term, no. 1, white side up; GL1: greenhouse film, long term, no. 1; * mean thickness±standard deviation measured at ten random points on a sample with an approximately 20 cm x 20 cm size; † manufacturer information.

Due to the limited number of samples, after the UV treatment, one sample of each mulch film was divided into two 2 cm x 2 cm pieces. One half was first used to perform the roughness test, then treated with mechanical abrasion and finally analysed for roughness, wettability, and chemical changes. The second half was only UV-treated and used to check the mechanical properties. The second sample of each mulch film kept its original size (2 cm x 4 cm); however, one-half was used to check for chemical change, and the other half was used for wettability change. The mechanical abrasion treatment was designed to simulate the potential mechanical abrasion that may occur due to wind-induced movement of the film on soil (particularly important for mulch films) and the possible impact of soil particles moved with the wind over the film surface (Fig 1). Therefore, the samples were mixed with 20 g of standard loamy sand soil (LUFA Speyer, 3.6±1% of clay, 8.6±1% of silty, 87.8±1% of sand) in a 60 mL amber screw neck glass vessel. The vessels were placed in a roller mixer (RG11, Roller Grill, France) with 4 rpm velocity for 61 days, with no replicates. The abrasion treatment was carried out in dry conditions at room temperature, without temperature control. After the abrasion exposure and before the material analysis, all abrasion samples were washed in an ultrasonic bath (130/300 W, 40 kHz) for 5 minutes using deionised water to remove excess soil particles.

### Analysing film properties

**Fourier transform infrared with attenuated total reflectance (FTIR-ATR).** FTIR-ATR was used to evaluate the chemical changes after the treatments for all samples. The technique is sensitive to changes in the polymer chemical

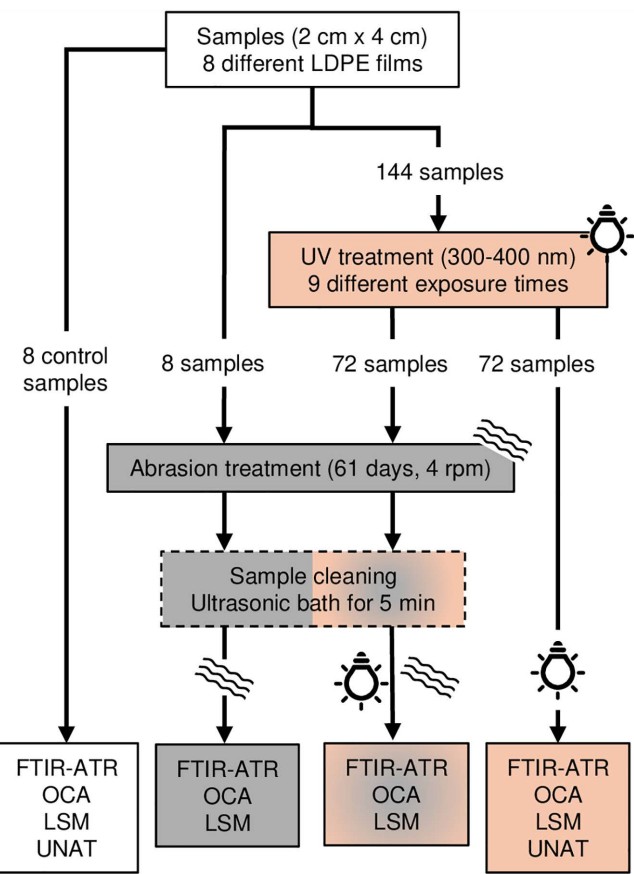

**Fig 1. Scheme of the sample treatment and analyses.**

structure and the detection of polar groups, such as C=O [24]. The chemical change was determined using an FTIR-ATR spectroscopy (Equinox 55, Bruker, USA) equipped with an ATR diamond crystal accessory with a penetration depth of around 2 μm.

Degradation was quantified by calculating the carbonyl index (CI) and the hydroxyl index (HI), which quantify the increase in carbonyl and hydroxyl groups, respectively, compared to a reference band. CI and HI were determined by dividing the area of the carbonyl band ($A_{carbonyl\ band}$), or the hydroxyl band, by the area of the reference band ($A_{reference\ band}$) (equation 1) [8, 24–27]. In addition to the area-based method, CI and HI were also calculated using the peak height values of the carbonyl (S1 Fig) and hydroxyl (S2 Fig) bands relative to a reference peak, allowing for a complementary evaluation of degradation. To calculate CI and HI, the wavelength range, as the wavenumber peaks, for the carbonyl, hydroxyl, and reference bands, was visually determined as given in Table 2

$$CI = \frac{A_{carbonyl\ band}}{A_{reference\ band}}$$

(1)

Each spectrum was collected in transmittance mode, from 4,000–400 $cm^{-1}$ spectral range, with 32 co-added scans per spectrum and 4 $cm^{-1}$ resolution. For each sample, three spectra were measured from random locations, while two spectra were collected on ML2B (treated with 52 days of UV exposure and abrasion) and MS1, MS2, ML1B, ML2B, ML1W, ML2W, and GL1 (treated with 80 days of UV exposure).

**Optical contact angle (OCA).** An OCA instrument (OCA35, DataPhysics, Germany) was used to measure the samples' wettability changes in sessile mode. Wettability usually measures the interaction between a solid and a liquid, where small angles (<90°C) refer to high wettability (hydrophilic) and high angles (>90°C) refer to low wettability (hydrophobic) [28]. The contact angle technique can detect the formation of polar groups on the surface of the photodegraded sample, which leads to an uneven charge distribution of the plastic surface that causes measurable differences in the interactions with water [24]. A 2 μL droplet of distilled water was applied at room temperature. Ten seconds after the application, the contact angle between the film sample and the water drop was measured 9 times. Each sample was measured at three randomly selected locations over the sample (3 locations 9 times, totalling 27 measurements).

**3D laser scanning confocal microscope (LSM).** The surface roughness of all samples was measured using an LSM (VK-X1000 Series, Keyence, Japan). The LSM was equipped with a 404 nm laser, and a confocal light was used to scan the sample surface before and after each test. The samples were placed on the same magnetic frame used for UV radiation exposure. Subsequently, high-resolution scans of the films were carried out with 500 μm magnification, obtaining for each sample a 1,024 x 768 raster with 2.7 μm x 2.7 μm cell size. The terrain ruggedness index (TRI) was

**Table 2. Carbonyl, hydroxyl, reference band range, and wavenumber peak used to calculate the carbonyl and hydroxyl indices.**

| Sample | Bands ($cm^{-1}$) | | | Peak ($cm^{-1}$) | | |
|---|---|---|---|---|---|---|
| | Carbonyl | Hydroxyl | Reference | Carbonyl | Hydroxyl | Reference |
| ML1B | 1,840–1,480 | 3,730–2,980 | 740–650 | 1,733 | 3,330 | 717 |
| ML2B | 1,840–1,480 | 3,730–2,980 | 740–650 | 1,733 | 3,330 | 717 |
| ML1W | 1,775–1,510 | 3,730–3,110 | 1510–1390 | 1,733 | 3,400 | 1,462 |
| ML2W | 1,775–1,510 | 3,730–3,110 | 1510–1390 | 1,733 | 3,400 | 1,462 |
| MS1 | 1,840–1,480 | 3,660–3,000 | 750–640 | 1,712 | 3,370 | 719 |
| MS2 | 1,840–1,480 | 3,660–3,000 | 750–640 | 1,712 | 3,370 | 719 |
| GL1 | 1,840–1,510 | 3,730–3,120 | 760–650 | 1,732 | 3,400 | 719 |
| GL2 | 1,840–1,510 | 3,730–3,120 | 760–650 | 1,732 | 3,400 | 719 |

calculated using the rasters in R software ('tri' function, 'spatialEco' library) to evaluate the mean small-scale roughness changes. The TRI function calculates the sum of the absolute differences in elevation between a specific raster cell and its surrounding cells [29]. In order to avoid smoothing effects by averaging over a large area, only the direct neighbouring cells of a 3 x 3 raster window were taken into account for the calculation of the TRI. Finally, the average TRI and its standard deviation were calculated over the entire area of each sample. Thereby, average surface roughness was determined using LSM, which integrates roughness across the full sample area to yield a representative mean for each treatment. This representative mean was prioritised over additional replicates of smaller areas.

**Universal nanomechanical tester (UNAT).** To measure changes in mechanical properties between pristine and UV-degraded samples, the variation in Young's modulus was recorded using a UNAT (ASMEC, Zwick/Roell, Germany). The UNAT applies controlled forces using a Berkovich indenter and records the material response, which allows for LDPE statements about crystallinity, brittleness, and elasticity [30]. The measurement is structured by gradually applying a force using a Berkovic indenter of 10 mN over 10 seconds (loading phase), maintaining this force for 5 seconds (holding phase), and then reducing the force to 0.260 mN over 4 seconds (unloading phase), with a fixed Poisson's ratio of 0.4. This procedure records the penetration depth and elastic and plastic deformation, allowing the obtention of sample elasticity [31]. We chose a subset of UV exposure encompassing the entire degradation spectrum of the different film types. The UV plus abrasion-treated samples were excluded from the analysis due to potential interference of soil particle attachment on the film surface. The following samples - MS2, ML1B, ML1W, and GL1, each subjected to UV treatment for 0, 21, 40, and 80 days – were selected and analysed by UNAT. The 0.5 cm x 1.5 cm samples were attached to glass slides with double-sided tape. The measurements were carried out 27 times at 3 different locations (position was randomly selected) on the sample. Within each location, 9 replicate measurements were conducted at close distances.

## Statistical analysis

All the data representing the different treatments, measurements and their residual values were tested for normality using a Kolmogorov-Smirnov normality test [32]. The samples analysed via FTIR-ATR and UNAT presented normally distributed data, the samples analysed via OCA presented a mixed distribution, and the samples analysed via LSM couldn't perform a normality test due to the lack of replicates. A regression line was performed to check for trends, an ANOVA test for the parametric data, and a Kruskal-Wallis test for the non-parametric data were performed to check for significant differences between treatments, using a significance level of $p < 0.05$ for all statistical tests. All statistical analyses and graphs were done using OriginPro 2023b (OriginLab, USA).

## Results

### Spectral characteristic and chemical groups

The spectral difference between the sample treated for 80 days and the pristine sample (Δ spectra) identifies the changes caused by the treatments (Fig 2). Although the Δ spectra highlight notable changes in the regions of 3,100–3,700 cm$^{-1}$ and 910–1,080 cm$^{-1}$ connected to the abrasion treatment (Fig 2b) [33,34], the figure may not be suitable for illustrating trends over time. As such, a dedicated index (i.e., carbonyl index) remains necessary to assess degradation over time and treatment.

The chemical analysis using the carbonyl index (area under the curve method) revealed no significant changes in the surface properties of the films subjected to both UV treatment and abrasion (Fig 3). This indicates that, for the major-ity of the samples, these treatments did not alter the chemical composition of the film surfaces. However, an exception was observed with the film sample MS2. This particular sample exhibited a significant change in its surface properties, demonstrating a clear linear relationship with the carbonyl index values ($r^2 = 0.80$ and 0.74). This suggests that the MS2 film is more susceptible to degradation under the combined effects of UV treatment and abrasion compared to the other samples. While for the height values, a significant change was observed for the samples MS2 ($r^2 = 0.69$ for the UV-treated

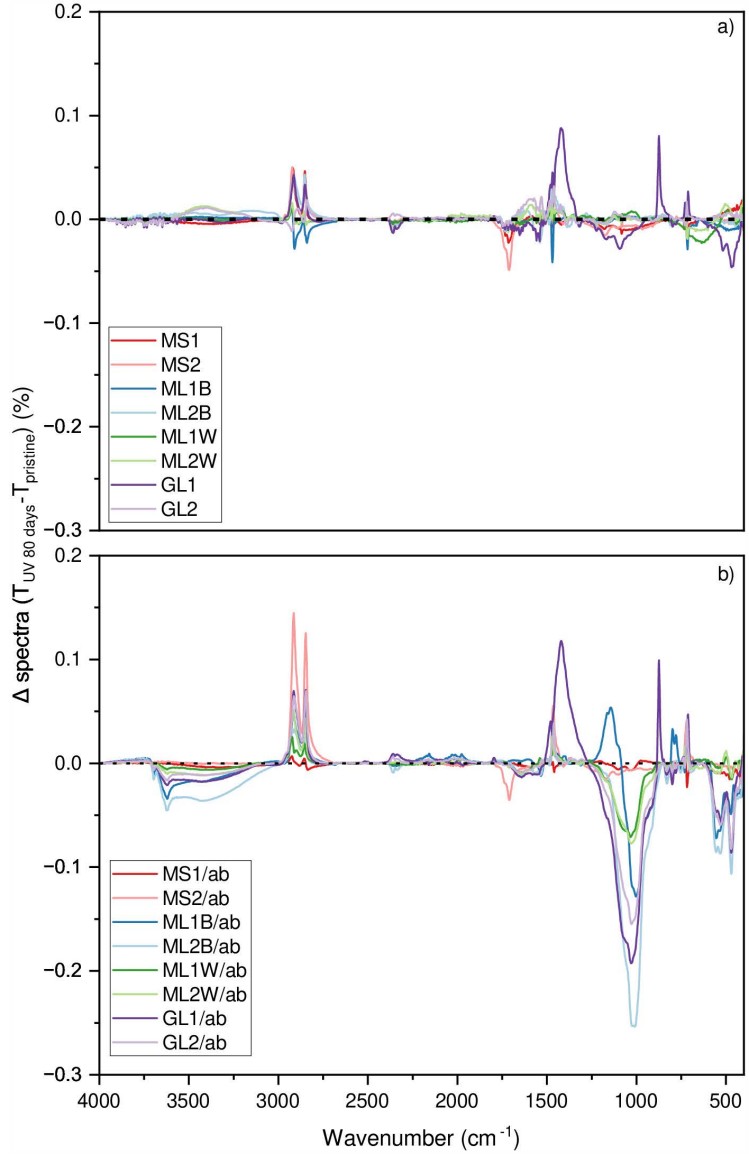

**Fig 2. Δ spectra between 80 days UV-exposed and pristine samples.** The different colours indicate the mean Δ spectra of the different samples tested in this study. Where a) represents the samples treated only with UV, and b) represents the samples treated with UV and abrasion. For an overview of the abbreviation, please see Table 1.

sample, $r^2 = 0.74$ for the UV plus abrasion-treated sample), ML1B ($r^2 = 0.51$ for the UV-treated sample), ML2B ($r^2 = 0.51$ for the UV-treated sample), GL1 ($r^2 = 0.49$ for the UV-treated sample), and GL2 ($r^2 = 0.40$ for the UV plus abrasion-treated sample) (S1 Fig).

The ANOVA test, for both methods, showed significant differences ($p < 0.05$) between the two treatments (UV vs. UV plus abrasion) for all samples. All samples showed significant differences when comparing different samples (e.g., MS1 and GL2) under the same treatment.

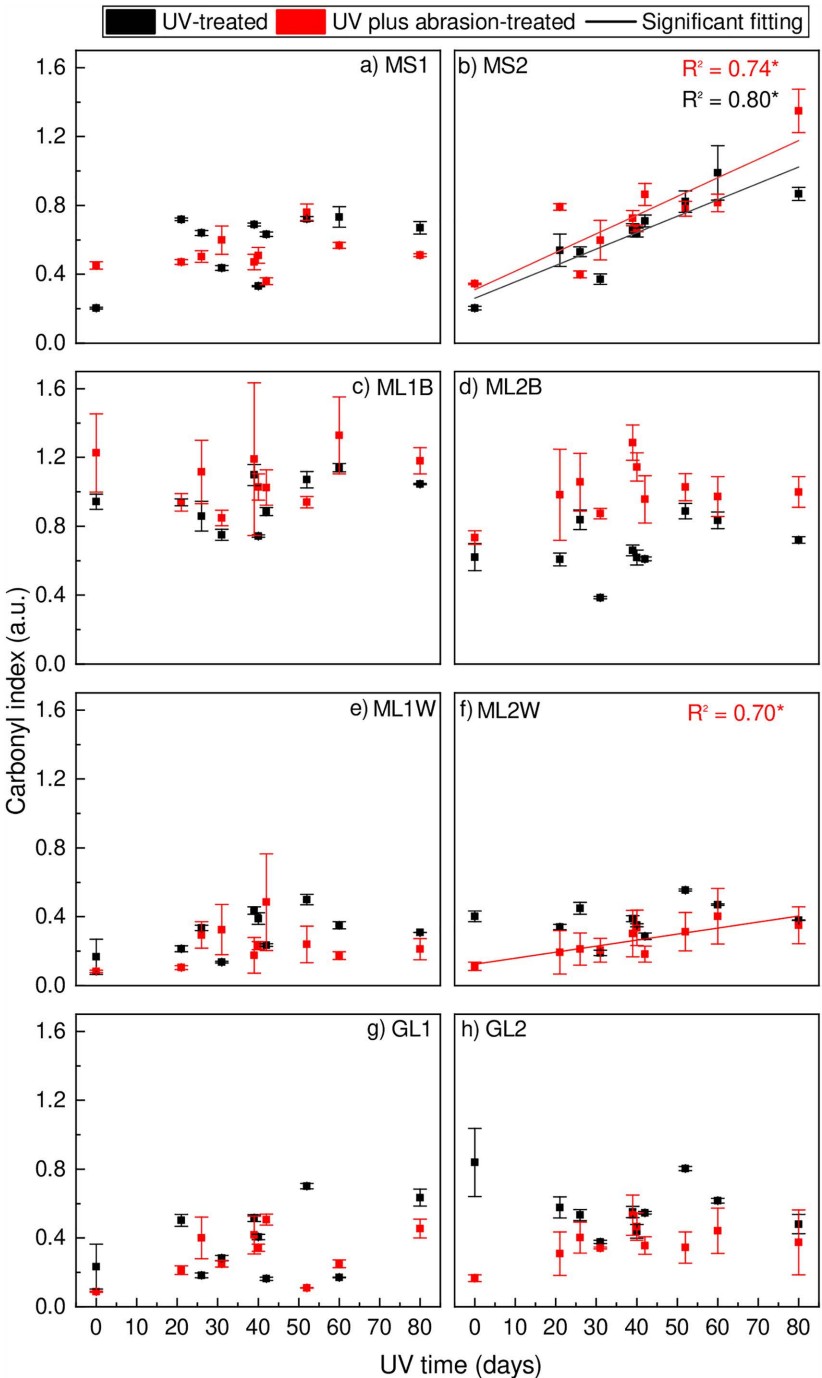

**Fig 3. Carbonyl index of a) MS1, b) MS2, c) ML1B, d) ML2B, e) ML1W, f) ML2W, g) GL1, and h) GL2.** *$p < 0.05$. For abbreviations, see Table 1.

Due to interferences caused by the abrasion test in the hydroxyl region (see Fig 2 and Table 2), the hydroxyl index using both methods (area under the curve (S3 Fig) and height value (S2 Fig)) was performed only for the UV-treated samples. However, none of the samples presented any significant change or linear trend.

## Wettability

The contact angle analysis of water droplets on the film surface (wettability) demonstrated a significant trend with UV exposure time for all UV-treated samples (Fig 4). The wettability of these samples increased rapidly from the pristine state to the UV-exposed state within the first 21 days of UV treatment. Following this initial surge, the rate of change in

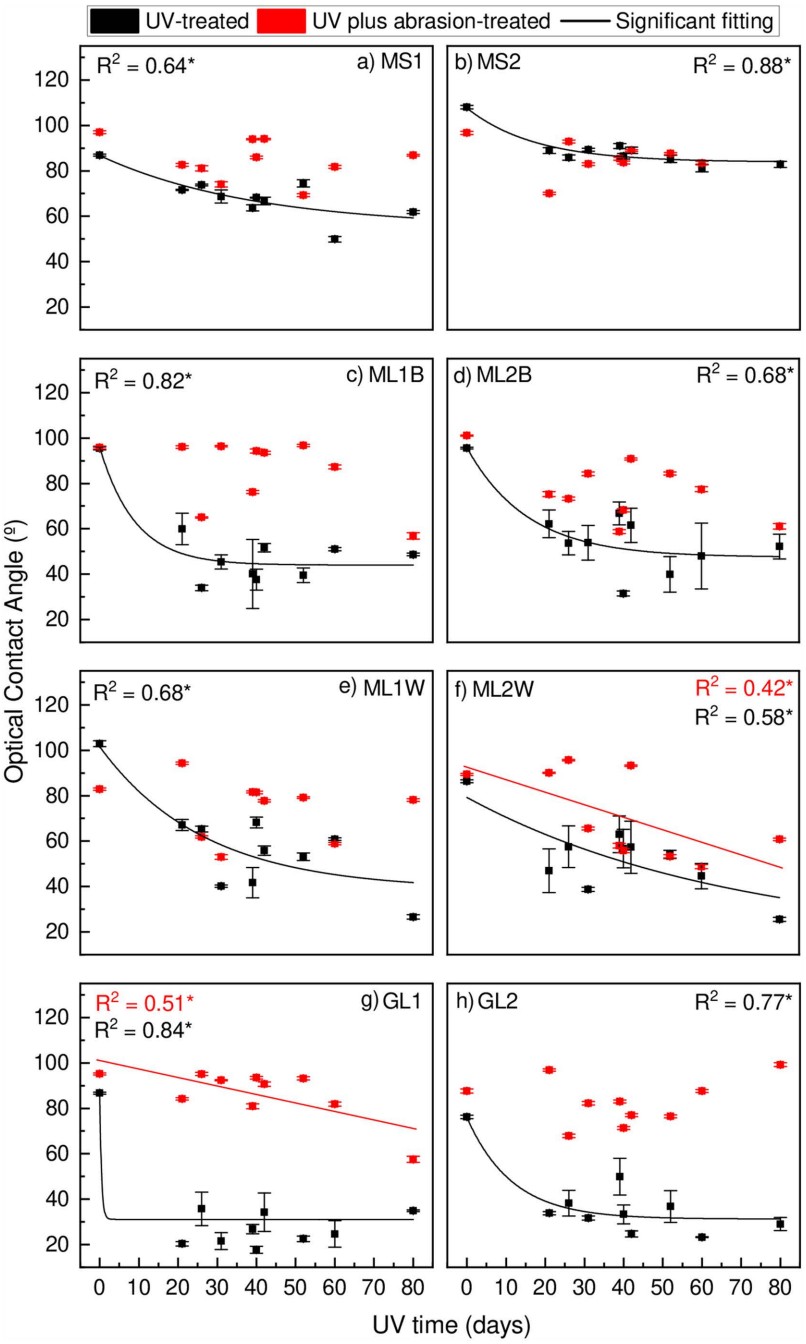

**Fig 4. Wettability of a) MS1, b) MS2, c) ML1B, d) ML2B, e) ML1W, f) ML2W, g) GL1, and h) GL2.** $*p < 0.05$. For abbreviations, see Table 1.

wettability slowed down for the remainder of the UV exposure period. The relationship between UV exposure time and wettability was exponential, with an r² value of at least 0.58. In contrast, the majority of samples treated with both UV and abrasion exhibited much higher wettability, similar to that of the pristine film, with no significant changes observed over the UV exposure time (Fig 4). The Kruskal-Wallis test indicated significant differences between the same samples when treated with different methods (UV vs. UV plus abrasion). Furthermore, samples with the same intended use but different manufacturers also showed significant differences.

## Surface roughness

The roughness analysis following UV treatment and abrasion revealed that almost all the UV plus abrasion-treated samples exhibited higher roughness compared to the samples treated only with UV, except for both greenhouse films (GL) samples (Fig 5g and h). Hardly any significant linear trend with UV radiation exposure time was observed for any of the samples except for the GL2 UV-treated sample (r²=0.44; Fig 5). An ANOVA test was performed to check the significant differences between different treatments for the same samples, and only the sample ML2W did not present a significant difference. Comparing different samples submitted to the same treatment, all samples presented significant differences.

## Elasticity

The elasticity test using the nanoindenter indicated no significant change in the Young's modulus (modulus of elasticity) for different UV radiation exposure amounts (Fig 6). However, was observed for the ML1B sample, a decrease from the pristine level to 21 days of UV exposure. It is important to note that the greenhouse film (GL1) exhibited much higher elasticity compared to the mulch films tested (ML1B and ML1W; Fig 6). The ANOVA test revealed that only the pristine ML1B sample showed a significant difference from the other ML1B samples tested. The changes in the elastic properties of the thin film sample (MS2) could not be measured due to technical limitations.

## Discussion

Various films used in plasticulture were exposed to different durations of UV radiation and to mechanical stress through mineral soil abrasion in this study. Subsequently, the films were assessed for changes in carbonyl and hydroxyl content, wettability, roughness, and elasticity. Even though two different methods for the degradation index are presented (area under the curve and height value), only the area under the curve method will be further discussed, as we believe this method increases the level of precision by considering all carbonyl species, series of overlaps, and band shifts present, avoiding error when choosing one specific peak for the height [26].

Both the FTIR-ATR-based index (carbonyl and hydroxyl) and the wettability of plastic samples are related to changes in polar groups [24,25,27]. Although the hydroxyl band could serve as an alternative assessment for degradation, the abrasion treatment introduced spectral interference in this region (Fig 2) [33]. Since the abrasion test interference makes it difficult to accurately compare the UV-treated and UV and abrasion-treated samples, the hydroxyl index will not be further discussed.

Abrasion could have introduced cracks that allowed oxygen to diffuse into the sample, potentially leading to observable degradation (Fig 3). Alternatively, the abrasion process might have removed the outer surface layer, exposing a more preserved inner part of the samples (Fig 3). While the range of UV exposure carried out within this study did not yield a pronounced significant change in the carbonyl index for the majority of samples (Fig 3), the wettability (reduction of water droplet contact angle) of the samples showed a substantial increase after a short exposure time (Fig 4).

The wettability increased rapidly after the first period of UV radiation exposure, but showed very limited reaction to additional UV radiation. Hence, wettability changes after a short sunlight exposure time, indicating the rapid formation of polar groups, which still fall under the detection limit of the FTIR-ATR-based carbonyl index (Fig 3). Based on Rouillon et al. [27], who used other analyses, mainly mechanical properties analyses, stated that the detection of early-stage

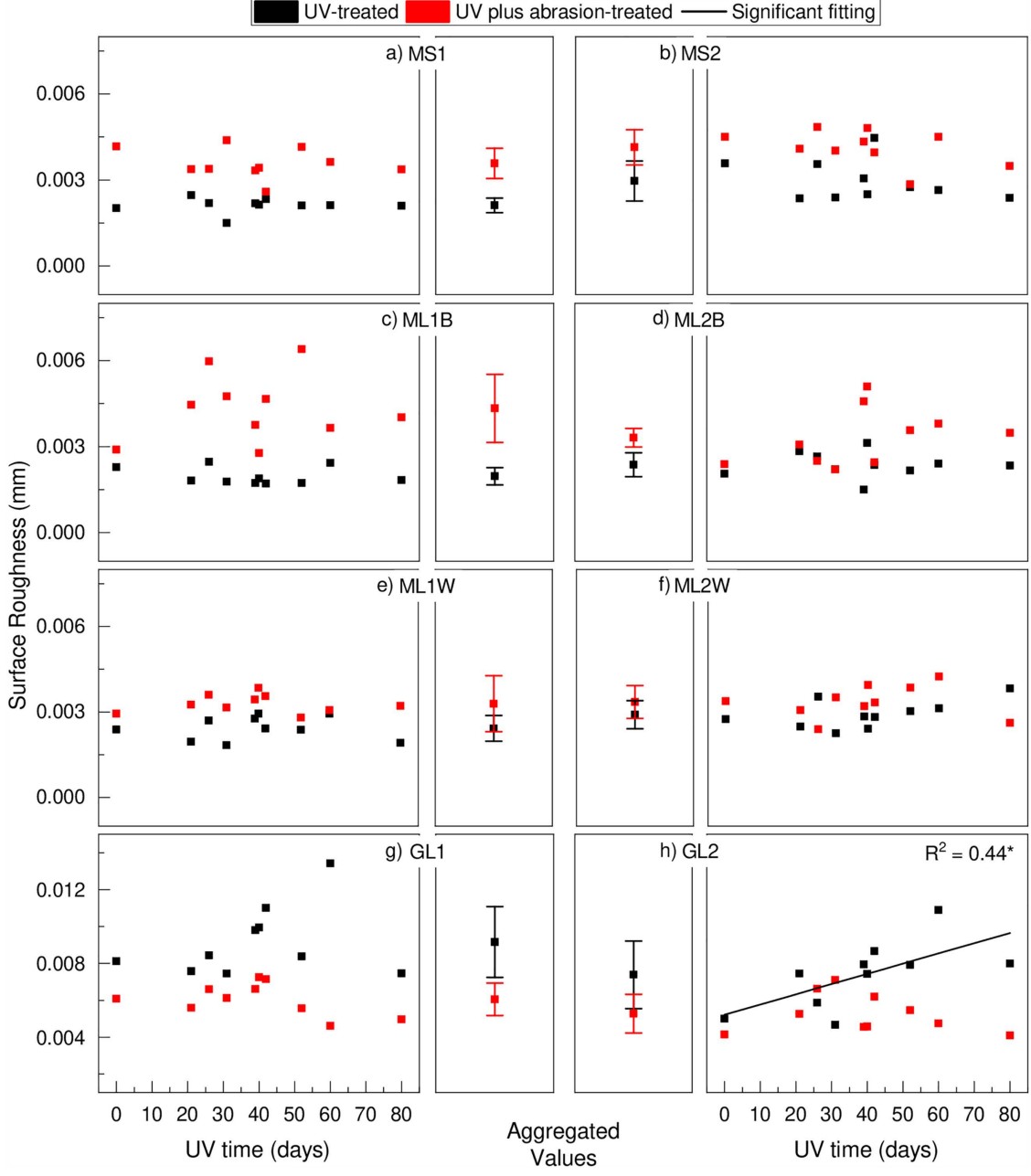

**Fig 5. Surface roughness of a) MS1, b) MS2, c) ML1B, d) ML2B, e) ML1W, f) ML2W, g) GL1, and h) GL2.** *p < 0.05. The samples GL1 and GL2 are presented in different scales for better visualisation. For abbreviations, see Table 1.

degradation of polypropylene could be possible even before the detection of carbonyl by FTIR. When similar samples are exposed to UV influence, transparent samples are expected to present a higher effect by the UV treatment (transparent > white > black). However, it needs to be highlighted that sample MS2 (mulch film designed for short-term application) showed an exceptionally high photodegradation response according to the carbonyl index (80% and 74% explained variance) in comparison to the sample MS1. A likely explanation is a substantially different plastic composition between polymers and additives (e.g., UV stabilisers) [7], which vary greatly among different producers and products and is a

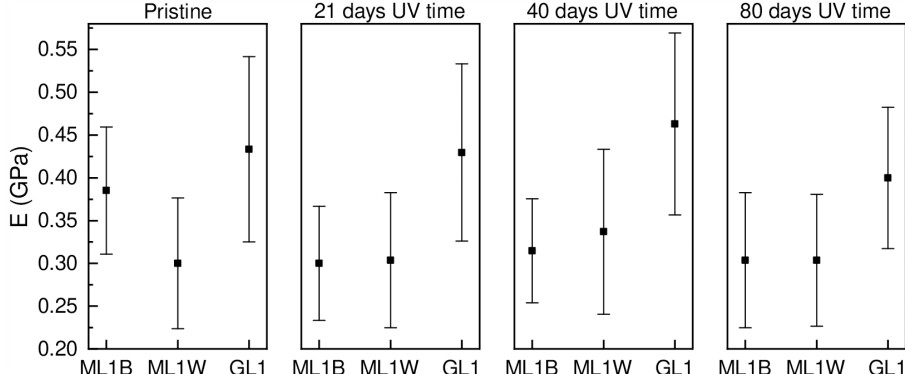

**Fig 6. Young's modulus (E) of UV-treated samples.** For abbreviations of the tested films, see Table 1.

closely guarded trade secret. The use of additives is crucial for the degradation processes. Nevertheless, the different concentrations and types of additives will have distinct effects, as the combination of different polymer types and additives may present different sensitivities, influencing the degradation mechanisms [35]. As shown in this study, films with similar purposes may react differently (Fig 3). For practitioners, the formation of microplastics (< 5 mm) and, thereby, plastic input to their soil systems can hardly be estimated, even though the film stability is tested with respect to the UV exposure and application time [11].

Using wettability for quantifying polar groups can serve as an early detection of degradation [25]. A large change in wettability by photodegradation was also observed by Suresh et al. [36] and Yang et al. [37]. Additionally, in line with our results, Liang et al. [38] also reported higher wettability on samples aged under soil burial. While wettability is also sensitive to changes in surface roughness [25,28], this study showed that surface roughness changed only when abrasion was applied (Fig 5). This indicates that UV radiation does not cause larger surface morphological changes, such as detachment of microplastics, that are detectable by the laser scanning microscope (2.7 μm x 2.7 μm raster size), while the abrasive force has sufficient energy to change the film surface morphology and thereby release plastic particles [18,19,21].

For all mulching films, the non-UV but abrasion-treated sample roughness is higher than the pristine film conditions (Fig 5), indicating that abrasion is not limited to degraded films. Interestingly, the pristine transparent greenhouse film (GL1 and GL2) is distinctively rougher compared to the mulch film and shows smoothening as a reaction to abrasion (Fig 5). Abrasion has been shown to increase the wettability of UV-treated films, which points to the removal of surface particles by abrasion. Hence, a destabilisation of the polymer structure by photodegradation makes films susceptible to detachment by abrasion [17–19,21,25]. Given the limited penetration depth of UV radiation and the associated formation of polar groups, abrasion can act as an agent to expose undegraded polymers to the surface and UV radiation. Hence, UV light and abrasion are interacting processes that can accelerate microplastic formation [18,19,21], with thinner films presenting a higher degradation rate [39,40]. Thereby, plasticultural regions where abrasion is a more frequent process, such as storm-prone regions, where phases of UV radiation exposure frequently alternate with abrasion, may experience accelerated microplastic formation as degraded polymers are removed from the surface and stop sheltering undegraded deeper-located polymers from UV radiation [8,24]. However, no microplastic analysis was performed to quantify the formation of microplastics.

Our results show that photodegradation requires very few days of accelerated UV radiation exposure and could potentially form detachable microplastic particles that abrasion processes in the soil system can release. Reduced film elasticity increases fragmentation to soils either during the application of mulch and greenhouse film or during the removal process after the cropping season. The UV radiation exposure time applied within this study (max. 80 days) did not cause changes

in the elasticity of the tested samples (Fig 6). Based on an estimated conversion from artificial UV exposure to natural UV exposure (see experimental design section), and only considering UV radiation influence, as weathering could be influenced by other factors such as temperature and humidity, all mulch films used in this study fall within the manufacturer's specified range of usability (Table 1). Indications for a loss of film stability were not identified after the applied UV radiation exposure time. However, the lack of substantial change in film elasticity in this study does not necessarily exclude fragmentation of mulch and greenhouse films over longer exposure times and/or combined environmental factors. It only highlights that the parameters used in this study were insufficient to cause fragmentation.

## Conclusion

In this study, different mulch and greenhouse films commonly used in plasticulture were exposed to various amounts of UV radiation and mechanical abrasion in a laboratory experiment. Even though the UV radiation exposure time for all films was less than the designated lifetime indicated by the manufacturer, our results show clear evidence of photodegradation on the different agricultural plastics used in this study, as well as an accelerating effect of abrasion on the degradation process. Additionally, we found indications of potential microplastic formation associated with abrasion processes. Mechanical abrasion of surface particles proved to be an efficient process for exposing non-degraded surfaces. These results suggest that the removal of surface particles by abrasion exposes undegraded film layers to UV radiation, effectively acting as a degradation accelerator. This study indicates that microplastic formation occurs almost immediately after application and that abrasion is an important modulator of the degradation rate. However, the films tested in our study showed no loss of elasticity and, therefore, no increase in fragmentation potential during agricultural management.

Future laboratory-based research should consider other environmental variables also important for degradation processes that were not considered in this study, such as humidity and temperature. A logical next step is to monitor UV-induced film degradation under real-world field conditions.

## Supporting information

**S1 Fig. Carbonyl index of the different mulch film UV-treated and UV plus abrasion-treated samples using the peak height values method.** For an overview of the abbreviation, please see Table 1 in the manuscript.
(TIF)

**S2 Fig. Hydroxyl index of the different mulch film UV-treated samples using the peak height values method.** For an overview of the abbreviation, please see Table 1 in the manuscript.
(TIF)

**S3 Fig. Hydroxyl index of the different mulch film UV-treated samples using the area under the curve method.** For an overview of the abbreviation, please see Table 1 in the manuscript.
(TIF)

## Acknowledgments

We want to thank the staff of the physics and materials engineering departments from the University of Augsburg, for providing the equipment needed and training, when necessary, to complete this study.

## Author contributions

**Conceptualization:** Ana Carolina Cugler Moreira, Florian Wilken, Peter Fiener.

**Formal analysis:** Ana Carolina Cugler Moreira, Alessandro Fabrizi.

**Funding acquisition:** Florian Wilken, Peter Fiener.

**Investigation:** Ana Carolina Cugler Moreira.

**Methodology:** Ana Carolina Cugler Moreira, Florian Wilken, Peter Fiener.

**Project administration:** Florian Wilken, Peter Fiener.

**Resources:** Florian Wilken, Peter Fiener.

**Supervision:** Florian Wilken, Peter Fiener.

**Validation:** Ana Carolina Cugler Moreira, Florian Wilken, Alessandro Fabrizi, Peter Fiener.

**Writing – original draft:** Ana Carolina Cugler Moreira.

**Writing – review & editing:** Ana Carolina Cugler Moreira, Florian Wilken, Alessandro Fabrizi, Peter Fiener.

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
