## [Decision Letter · Decision Letter 0]

29 Jan 2026

Dear Dr. Fiener,

Thank you for submitting your manuscript to PLOS ONE. After careful consideration, we feel that it has merit but does not fully meet PLOS ONE’s publication criteria as it currently stands. Therefore, we invite you to submit a revised version of the manuscript that addresses the points raised during the review process.

We look forward to receiving your revised manuscript.

Kind regards,

Phuping Sucharitakul

Academic Editor

PLOS One

Journal Requirements:

“European Union’s Horizon 2020 research and innovation programme under the Marie Skłodowska-Curie grant agreement No 955334.”

5. We notice that your supplementary figures are uploaded with the file type 'Figure'. Please amend the file type to 'Supporting Information'. Please ensure that each Supporting Information file has a legend listed in the manuscript after the references list.

Reviewers' comments:

Reviewer's Responses to Questions

**Comments to the Author**

1. Is the manuscript technically sound, and do the data support the conclusions?

Reviewer #1: Yes

Reviewer #2: Yes

2. Has the statistical analysis been performed appropriately and rigorously?

Reviewer #1: Yes

Reviewer #2: Yes

3. Have the authors made all data underlying the findings in their manuscript fully available?

Reviewer #1: Yes

Reviewer #2: Yes

4. Is the manuscript presented in an intelligible fashion and written in standard English?

Reviewer #1: Yes

Reviewer #2: Yes

Reviewer #1: General Comments

This is an interesting manuscript that addresses a relevant topic, and the technical aspects are sound, and the interpretations presented by the authors are generally convincing. The overall aim of the study and its main findings are clearly conveyed, and the topic itself has sufficient relevance for the readership of the journal. While the level of novelty is moderate, the work can still make a meaningful contribution provided that the presentation and technical details are carefully refined.

The overall structure of the manuscript is reasonable, with a logical flow from introduction to conclusion. The methodology appears appropriate for addressing the stated research questions, and no fundamental methodological flaws are evident based on the current description. The discussion largely reflects the results, although clarity and readability could be improved in several places. The conclusion section would benefit from a clearer synthesis of the main findings, along with a discussion of limitations and directions for future research.

At this stage, I believe the manuscript is potentially suitable for publication after revision. The requested changes are mainly related to consistency, formatting, clarity, and completeness, but addressing them carefully will significantly improve the quality and professionalism of the manuscript.

Specific Comments

L4: Please check whether the reference numbering style used here is appropriate for this journal. If references are not meant to be indicated as superscripts, then this numbering style is acceptable, but it must be applied consistently throughout the entire manuscript.

L10: When the same unit is repeated for a series of values, it is sufficient to indicate the unit only once at the end of the series. Please revise accordingly.

L57: Please specify where the material or equipment was purchased. The location and the company name should be clearly stated.

L60: When indicating numerical ranges, please use an en dash instead of a hyphen.

L66, L67: Please revise “meter” to the SI unit symbol “m” and ensure consistent usage throughout the manuscript.

L82: The dates listed here appear rather arbitrary. Please clarify the rationale or criteria used for selecting these particular dates.

L125: Please insert commas for numbers at the thousand level (e.g., 1,000) and apply this formatting consistently throughout the manuscript.

L136: For temperature units in °C, the unit is usually written immediately after the number without a space. If this style is adopted, please ensure consistency across the entire manuscript.

L190, L227, L243, L257: Please remove the underlining from the titles and revise them to conform to the journal’s standard format for chapters and section headings.

L193: Please use an en dash for numerical ranges and insert commas for numbers at the thousand level.

L270: The paragraphs in the discussion section are excessively long. For better readability, please divide them into shorter paragraphs based on their logical and thematic structure.

L341: The conclusion should more clearly summarize the main findings of the study. In addition, it would be beneficial to explicitly mention the limitations of the current work and to suggest directions for future research. Please revise this section from that perspective.

L373: Overall, the number of references cited in the manuscript is relatively small. Please consider strengthening the literature review by incorporating additional relevant and up-to-date references.

Reviewer #2: 1. The manuscript repeatedly refers to “microplastic formation” and “release of microplastic particles,” yet no direct quantification or identification of microplastics (e.g., particle counting, size distribution, mass balance) is performed. The conclusions are instead inferred from changes in surface roughness, wettability, and FTIR indices. While this interpretation is reasonable, the wording in the Discussion and Conclusion sections should be softened. Statements implying confirmed microplastic generation should be reframed as potential or indirect evidence of microplastic formation unless supported by direct measurements.

2. The abrasion treatment was performed without replicates (61 days, single vessel per treatment). Given the variability inherent to abrasion processes, this limits the statistical robustness of the results, particularly for surface roughness and FTIR-based indices. This limitation should be explicitly acknowledged in the Methods and Discussion sections. The authors may also consider clarifying why replication was not feasible and how this might affect result interpretation.

3. The manuscript attempts to relate artificial UV exposure to natural sunlight exposure using a generalized energy conversion approach. While helpful, this conversion remains approximate and does not account for additional environmental factors such as temperature fluctuations, moisture, or soil-film contact. The Discussion would benefit from a clearer statement that the laboratory exposure represents a simplified scenario and may not fully capture field-scale degradation dynamics.

4.The conclusion that no increased fragmentation potential exists is based on unchanged Young’s modulus values within the applied UV exposure range. However, fragmentation in real agricultural systems is influenced not only by elastic modulus but also by embrittlement, crack initiation, and handling stresses. The authors should clarify that the absence of elasticity changes under the tested conditions does not necessarily rule out fragmentation under longer exposure times or combined environmental stresses.

5.The terms “macroplastic residues,” “plastic particles,” and “microplastic formation” are sometimes used interchangeably. A short clarification of terminology early in the manuscript would improve readability.

6. The manuscript notes that differences between films may arise from undisclosed additives. While this is reasonable, a brief discussion of how this uncertainty affects comparability between film types would strengthen the interpretation.

**Do you want your identity to be public for this peer review?** For information about this choice, including consent withdrawal, please see our Privacy Policy

Reviewer #1: No

Reviewer #2: No

---

## [Author Response · Author response to Decision Letter 1]

20 Feb 2026

The reviewers' comments are addressed in the "Response to the reviewers" file.

---

## [Decision Letter · Decision Letter 1]

26 Feb 2026

UV light and abrasion’s role in degrading plasticulture films

PONE-D-25-56979R1

Dear Dr. Fiener,

We’re pleased to inform you that your manuscript has been judged scientifically suitable for publication and will be formally accepted for publication once it meets all outstanding technical requirements.

Kind regards,

Phuping Sucharitakul

Academic Editor

PLOS One

Additional Editor Comments (optional):

Reviewers' comments:

Reviewer's Responses to Questions

**Comments to the Author**

Reviewer #1: All comments have been addressed

Reviewer #2: All comments have been addressed

2. Is the manuscript technically sound, and do the data support the conclusions?

Reviewer #1: Yes

Reviewer #2: Yes

3. Has the statistical analysis been performed appropriately and rigorously?

Reviewer #1: Yes

Reviewer #2: Yes

4. Have the authors made all data underlying the findings in their manuscript fully available?

Reviewer #1: Yes

Reviewer #2: Yes

5. Is the manuscript presented in an intelligible fashion and written in standard English?

Reviewer #1: Yes

Reviewer #2: Yes

Reviewer #1: The authors have revised the manuscript; however, I would like to request several additional corrections as outlined below:

L4: Please format the references using bracketed numbering (e.g., [1]) instead of parentheses.

L42: The citation format for Ren and Ni (2022) is incorrect. In addition, it appears again in L44. Please revise accordingly.

L75: 1800 W → 1,800 W

L199: 3,100–3,700 cm⁻¹ and 910–1,080 cm⁻¹ → 3,100–3,700 and 910–1,080 cm⁻¹

L214: The term r² is used without prior definition. Please define it before its first occurrence in the text.

Reviewer #2: (No Response)

**Do you want your identity to be public for this peer review?** For information about this choice, including consent withdrawal, please see our Privacy Policy

Reviewer #1: No

Reviewer #2: No

---

## [Editor Report · Acceptance letter]

PONE-D-25-56979R1

PLOS One

Dear Dr. Fiener,

I'm pleased to inform you that your manuscript has been deemed suitable for publication in PLOS One. Congratulations! Your manuscript is now being handed over to our production team.

Kind regards,

on behalf of

Dr. Phuping Sucharitakul

Academic Editor

PLOS One